# Histone Acetylation Domains Are Differentially Induced during Development of Heart Failure in Dahl Salt-Sensitive Rats

**DOI:** 10.3390/ijms22041771

**Published:** 2021-02-10

**Authors:** Masafumi Funamoto, Yoichi Sunagawa, Yasufumi Katanasaka, Kana Shimizu, Yusuke Miyazaki, Nurmila Sari, Satoshi Shimizu, Kiyoshi Mori, Hiromichi Wada, Koji Hasegawa, Tatsuya Morimoto

**Affiliations:** 1Division of Molecular Medicine, School of Pharmaceutical Sciences, University of Shizuoka, 52-1 Yada, Suruga-ku, Shizuoka 422-8526, Japan; funamoto@u-shizuoka-ken.ac.jp (M.F.); y.sunagawa@u-shizuoka-ken.ac.jp (Y.S.); katana@u-shizuoka-ken.ac.jp (Y.K.); s18804@u-shizuoka-ken.ac.jp (K.S.); y.miyazaki@u-shizuoka-ken.ac.jp (Y.M.); nurmilasari@gmail.com (N.S.); s18410@u-shizuoka-ken.ac.jp (S.S.); koj@kuhp.kyoto-u.ac.jp (K.H.); 2Kyoto Medical Center, Clinical Research Institute, National Hospital Organization, 1-1 Fukakusa Mukaihatacho, Fushimi-ku, Kyoto 612-8555, Japan; hwada@kuhp.kyoto-u.ac.jp; 3Shizuoka General Hospital, 4-27-1 Kitaando, Aoi-ku, Shizuoka 420-8527, Japan; kiyoshimori2001@gmail.com

**Keywords:** p300, histone, acetylation, BRG1, H3K9, H3K122, cardiac hypertrophy, heart failure

## Abstract

Histone acetylation by epigenetic regulators has been shown to activate the transcription of hypertrophic response genes, which subsequently leads to the development and progression of heart failure. However, nothing is known about the acetylation of the histone tail and globular domains in left ventricular hypertrophy or in heart failure. The acetylation of H3K9 on the promoter of the hypertrophic response gene was significantly increased in the left ventricular hypertrophy stage, whereas the acetylation of H3K122 did not increase in the left ventricular hypertrophy stage but did significantly increase in the heart failure stage. Interestingly, the interaction between the chromatin remodeling factor BRG1 and p300 was significantly increased in the heart failure stage, but not in the left ventricular hypertrophy stage. This study demonstrates that stage-specific acetylation of the histone tail and globular domains occurs during the development and progression of heart failure, providing novel insights into the epigenetic regulatory mechanism governing transcriptional activity in these processes.

## 1. Introduction

Heart failure (HF) is a major health problem and a leading cause of death worldwide [1,2]. Recently, epigenetic regulatory mechanisms, including histone post-translational modification, have attracted attention as key processes in HF development [3,4,5]. The main types of post-translational histone modification are acetylation and methylation. These modifications play essential roles in a variety of regulatory mechanisms and are controlled by acetyltransferase or methylase [6]. The roles of histone acetyltransferases (HATs) and histone deacetylases in HF have been intensively studied [7,8,9]. Wei et al. reported that the expression level of the histone acetyltransferase p300 was significantly increased in the hearts of patients with dilated cardiomyopathy or ischemic HF compared to that of healthy individuals [10]. In the development of HF, p300 promotes the expression of atrial natriuretic factor (ANF), brain natriuretic peptide (BNP), and β-myosin heavy chain (β-MHC) by acetylating the transcription factors GATA4 and MEF2 [9,10]. Previous research using transgenic (TG) mice that exhibited cardiac-specific p300 overexpression showed that the heart gradually enlarged, leading to heart failure, and that left ventricular remodeling after myocardial infarction was greater in these mice than in wild-type (WT) mice [9]. However, in TG mice that exhibited cardiac-specific overexpression of mutant p300 lacking HAT activity, the exacerbation of left ventricular remodeling after myocardial infarction was attenuated to the same extent as in WT mice [8,9]. Conversely, it has been reported that homozygous knockout of p300 causes embryonic lethality in mice and that heterozygous knockout of p300 suppresses left ventricular hypertrophy (LVH) induced by transverse aortic coarctation (TAC) surgery in mice [10,11]. The administration of curcumin, a natural p300-specific HAT inhibitor, both to rats with myocardial infarction and to hypertensive heart disease model rats has been shown to suppress the development of HF [12]. In addition, anacardic acid, which is a natural p300 HAT inhibitor, also suppressed cardiac hypertrophy and acetylation of the histone tail domain H3K9 in mice given TAC surgery [13]. Furthermore, the p300 HAT-specific inhibitors C646 and L002 suppress left ventricular wall thickening induced by angiotensin II treatment [14]. Taken together, the above findings suggest that p300 HAT activity is a potential target for HF treatment.

HATs such as p300 accelerate gene transcription via epigenetic mechanisms, including chromatin remodeling, in which chromatin changes structurally from heterochromatin to euchromatin [15,16,17]. The basic unit of chromatin, the nucleosome, is composed of 147 bp of DNA wrapped around the histone octamer [18]. The histone tail domain has several basic amino acid residues that are subject to modification [19]. More than 150 of these modifications have been reported, including acetylation, methylation, phosphorylation, SUMOylation, and ubiquitination [20]. Acetylation of the histone tail domain has been intensively studied, including the domains H3 lysine 9 (H3K9) and 14 (H3K14) [21,22,23,24]. Transcriptional regulators expose DNA and interact with transcription factors by congregating at the acetylation site of the histone tail domain. As a consequence, these transcriptional regulators facilitate the recruitment of the transcription factors to the DNA. In other words, the acetylation of the histone tail domain plays an important role in chromatin remodeling [25]. However, there have been a few studies on the acetylation of the globular domain. Recently, H3 lysine 122 (H3K122) has been reported to be a novel p300 histone acetylation site. H3K122 is located on a globular domain on the surface of the histone, which is wrapped in DNA. H3K122 acetylation attenuates histone-DNA binding and dramatically activates gene transcription [26]. That is, H3K122 acetylation induces the process known as nucleosome remodeling, in which the nucleosome relaxes and DNA is exposed, to a greater extent than the acetylation of the tail domain. The acetylation of the histone globular domain directly affects histone–DNA interaction and plays an important role as a platform in nucleosome remodeling.

Epigenetic regulatory mechanisms such as histone post-translational modifications are involved in physiological functions and in the development of many diseases [27,28,29]. Studies of histone acetylation levels in HF carried out with chromatin-immunoprecipitation (ChIP)-sequencing using HF mice that had undergone TAC surgery have shown that the acetylation of H3K9 and H3K27 located in the histone tail domain is enhanced in the entire gene region by p300 [30]. However, nothing is known about the acetylation of the histone globular domain in HF with LV systolic dysfunction. Therefore, the purpose of the present study is to examine how the acetylation of the histone tail and globular domains changes during the transition from the LVH to HF stages in Dahl rats, a model of hypertension-induced HF. The results reveal for the first time that, in the LVH stage, the acetylation of the tail domain (H3K9) is enhanced but that of the globular domain (H3K122) is not and that, in the HF stage, the acetylation of H3K122 is enhanced. The results also reveal that the formation of a complex that combines p300 with the chromatin remodeling factor BRG1 occurs at a greater degree during the transition from the LVH stage to the HF stage. These findings suggest that the histone acetylation domain changes from the tail domain to the globular domain during the transition from the LVH stage to the HF stage and that the formation of the p300/BRG1 complex is involved in this change in the histone acetylation domain. These novel findings on the involvement of epigenetic regulatory mechanisms in the development of heart failure may open new paths to the treatment of the disease.

## 2. Results

### 2.1. Acetylation Levels of H3K9 and H3K122 Were Elevated by Hypertrophic Stimulation in Cardiomyocytes

To investigate whether H3K9 and H3K122 are acetylated by hypertrophic stimulation in primary cardiomyocytes, cardiomyocytes were stimulated with saline or 30 μM phenylephrine (PE), an α1-adrenergic agonist, for 48 h. Protein extracts from these cells were subjected to immunoblotting with the antibodies against the acetylated forms of H3K9 or H3K122. The results showed that acetylation levels of H3K9 and H3K122 were increased by the stimulation with PE (Figure 1A). Nuclear hyperacetylation was evaluated by immunostaining with anti-acetyl-H3K9 or anti-acetyl-H3K122 antibody. PE stimulation markedly induced nuclear staining, indicating the hyperacetylation of H3K9 and H3K122 (Figure 1B). To assess the accumulation levels of acetylated H3K9 and H3K122 around the promoters of hypertrophic response genes, a ChIP assay was performed. The acetylation levels of H3K9 and H3K122 at the ANF, BNP, and β-MHC gene promoters were significantly increased at 4 h after PE stimulation (Figure 1C–H). On the other hand, the acetylation levels of H3K9 and H3K122 around the upstream region of these promoters were not changed after PE stimulation (Appendix A).

### 2.2. P300-HAT Inhibition Suppressed H3K9 Acetylation, H3K122 Acetylation, and Cardiomyocyte Hypertrophy

Although H3K122 has been reported to be acetylated by p300, a histone acetyltransferase, in vitro [26], various types of HAT are present in cells, and the type of HAT differs depending on the cell type and the histone acetylation site [31,32,33]. Previous studies have found that p300 is required for acetylation and for the transcriptional activity of GATA4, as well as for pathological left ventricular hypertrophy and the development of HF in vivo [9,34,35]. To investigate whether p300 acetylates H3K9 and H3K122 in cardiomyocyte hypertrophy, cardiomyocytes were treated with p300 small interfering RNA (siRNA) or with curcumin, a p300 specific HAT inhibitor, and then stimulated with PE. Results of immunostaining and surface area measurement showed that p300 knockdown and curcumin suppressed PE-induced cardiomyocyte hypertrophy (Figure 2A,B and Appendix A). PE-induced increase in the mRNA levels of ANF, BNP, and β-MHC was suppressed by p300 knockdown and curcumin (Figure 2C,E and Appendix A). Moreover, both p300 knockdown and curcumin suppressed the acetylation levels of H3K9 and H3K122 by PE (Figure 2F–K). These results indicate that the HAT activity of p300 acetylates H3K9 and H3K122 in hypertrophic response.

### 2.3. Overexpression of P300 Enhanced H3K9 and H3K122 Acetylation

To investigate whether p300 overexpression increases the acetylation levels of H3K9 and H3K122, cardiomyocytes were transfected with pCMV-p300 CHA or pcDNA null vector. The results showed that p300 overexpression induced an increase in the acetylation levels of both H3K9 and H3K122 (Figure 3A–C). To investigate whether p300 overexpression also increases the acetylation levels of H3K9 and H3K122 in vivo, transgenic mice that exhibited cardiac-specific p300 overexpression (α-MHC-p300-TG mice) were used. This p300 overexpression has been shown to induce left ventricular remodeling and HF [7]. The heart weight/body weight ratio was significantly increased in the p300-TG mice compared to the wild-type (WT) mice at 26 weeks of age (Appendix A). As shown in Figure 3D–F, mRNA (ANF, BNP, and β-MHC) levels were significantly increased in the p300-TG mice compared to the WT mice. Moreover, WB analysis showed that the acetylation levels of H3K9 and H3K122 were also significantly increased in the p300-TG mice compared to the WT mice (Figure 3G). Next, an in vivo ChIP assay revealed that H3K9 and H3K122 were highly acetylated on the ANF, BNP, and β-MHC promoters in vivo (Figure 3H–M). These results indicate that p300 acetylates H3K9 and H3K122 in the heart in vivo.

### 2.4. H3K122 Acetylation Was Enhanced in Heart Failure

The above experiments demonstrated that the tail (H3K9) and globular (H3K122) domains of histone were acetylated by p300 in cardiomyocyte hypertrophy in vitro and in vivo. To investigate whether these histone acetylation levels increase around ANF, BNP, and β-MHC promoters in the HF stage, WB and in vivo ChIP assays were carried out with Dahl salt-sensitive (DS) rats, a hypertensive HF model. Echocardiography showed that left ventricular posterior wall thickness (LVPWT) was significantly increased in 12-week-old DS rats with preserved fractional shortening (FS) compared with 12-week-old Dahl salt-resistant (DR) rats with preserved FS. FS was significantly decreased in the 21w DS rats compared with the 21w DR rats (Appendix A). These results indicate that the 12w DS rats were at the LVH stage, in which systolic function is preserved, while the 21w DS rats were at the HF stage, in which systolic dysfunction is obvious. As shown in Figure 4A–C, the mRNA levels of ANF, BNP, and β-MHC were significantly increased in the 12w DS rats compared with the 12w DR rats. In addition, these mRNA levels were more significantly increased in the 21w DS rats compared with the 12w DS rats. WB analysis showed that H3K9 acetylation was increased in the 12w DS rats compared with the 12w DR rats, while H3K122 acetylation was increased in the 21w DS rats compared with the 21w DR rats but was not increased in the 12w DS rats (Figure 4D–F). To determine whether the acetylation levels of H3K9 and H3K122 changed around the ANF, BNP, and β-MHC promoters between the LVH stage and the HF stage, the heart lysates from the rats were subjected to an in vivo ChIP assay. The results showed that H3K9 acetylation in the heart was increased around the ANF, BNP, and β-MHC promoters in the 12w DS rats compared with the 12w DR rats. In contrast, H3K122 acetylation in the heart was increased around these promoters in the 21w DS rats compared with the 21w DR rats but not increased in the 12w DS rats (Figure 4G–L). These results demonstrate that the site of histone acetylation changes from the histone tail domain to the globular domain during the transition from the LVH stage to the HF stage.

### 2.5. Interaction between P300 and BRG1 Was Enhanced, and the Recruitment of BRG1 Was Increased in Heart Failure

The above experiments demonstrated that the increase in transcription that occurs with the transition from the LVH stage to the HF stage is associated with differences in the site of histone acetylation during each stage. To investigate whether the change in recruitment of p300 onto the reactive gene promoters affects this change in the histone acetylation domain, an in vivo ChIP assay was performed. The results showed that the amount of p300 recruited onto the ANF, BNP, and β-MHC promoters was much higher in the 12w and 21w DS rats compared with the 12w and 21w DR rats, respectively. However, the amount of p300 recruited was similar in the 12w DS and 21w DS rats (Figure 5A–C). This suggests that the change of histone acetylation sites between LVH and HF is not depended on the alternation of p300 recruitment to the promoter. Thus, we hypothesized that this change is regulated by other factors, such as chromatin remodeling factors. One factor that may be involved is BRG1. BRG1 is a catalytic subunit of the SWI/SNF chromatin remodeling complex, and it has been shown to cooperate with p300 in histone acetylation (H3K27) during transcriptional upregulation [36]. It is possible that BRG1 is associated with the change in sites of histone acetylation. Therefore, to investigate whether the amount of binding between BRG1 and p300 changes in the HF stage, an IP-WB assay was carried out at the HF and LVH stages. The results showed that the amount of this binding was increased in the 21w DS rats compared with the 21w DR rats but was not increased in the 12w DS rats (Figure 5D–H). To investigate whether BRG1 is associated with the change in the acetylation site from the tail domain to the globular domain during the transition from the LVH stage to the HF stage, an in vivo ChIP assay was performed. The results showed that the recruitment of BRG1 to the ANF, BNP, and β-MHC promoters was increased in the 21w DS rats compared with in the 21w DR rats, but not in the 12w DS rats (Figure 5I,J).

## 3. Discussion

Epigenetic regulation of HF has been intensively studied [37,38,39,40]. Building on these previous studies, the present study using Dahl rats, a model of hypertension-induced HF, has revealed for the first time not only that the acetylation of the histone tail domain H3K9 increases in the LVH stage but also that the acetylation of the histone globular domain H3K122 increases in the HF stage. The present study proposes a novel model of transcriptional regulation by stepwise acetylation of the histone tail domain and the globular domain during the transition from the LVH stage to the HF stage.

Histone post-translational modifications are involved in cellular processes such as transcription, replication, and DNA repair [41,42,43]. These processes increase the accessibility of transcription factors to DNA via chromatin remodeling [44,45]. Most histone post-translational modifications occur on histone tails and are induced by many factors, such as transcription factors and transcriptional regulators [46,47]. In transcription, this acetylation of the tail domain indirectly affects chromatin structure but not directly affects nucleosome structure. On the other hand, the histone globular domain H3K122 is specifically located on the DNA-binding surface of the histone body in the nucleosome structure. Tropberger et al. revealed that acetylated H3K122 reduces DNA–histone affinity and relaxes the nucleosome structure [48]. This and other studies also indicate that acetylated H3K122 has a direct effect on changes in nucleosome structure and that the acetylation of the tail and globular domains have different transcriptional regulatory mechanisms [25,26,48]. Fenley et al. demonstrated that the acetylation of only one lysine residue on the histone globular domain substantially reduced DNA–histone interaction [49]. However, the acetylated tail domain does not affect changes in the nucleosome structure [25]. H3K122 acetylation occurs frequently in gene promoter and enhancer regions where gene expression is upregulated. The levels of gene expression are directly increased by this acetylation [26]. In the present study, ANF, BNP, and β-MHC mRNA levels were significantly increased in the HF stage compared with the LVH stage. Acetylation of H3K122 in these gene promoter regions triggered explosive progress in transcription. Histone acetylation is the final common pathway in the signaling pathways of gene expression in the HF stage. This study suggests that H3K122 acetylation is the most significant trigger of the HF stage.

BRG1 is a chromatin remodeling factor and a major protein in the SWI/SNF complex [50,51,52,53]. It relaxes nucleosome structure in an ATP-dependent manner and enhances gene transcription [54,55], and it regulates cardiomyocyte development and differentiation by interacting with cardiac transcription factors such as GATA4, Nkx2-5, and TBX5 [56]. BRG1 is highly expressed during the fetal period, is down-regulated in neonatal hearts, and is highly expressed together with β-MHC in human cardiomyopathy. In BRG1 conditional knockout mice, HF induced by TAC surgery is suppressed compared with WT mice [54]. In embryonic stem cells, Brd4, a bromodomain protein, mediates histone H3 acetylation and chromatin remodeling via the formation of the BRG1/p300 complex [36]. In cardiac hypertrophy, it has been reported that the interaction of the Swi/Snf (Brm, BRG1) complex with HADAC2 and p300 is associated with the MHC isoform switch [57]. In addition, this switch has been shown to be regulated by epigenetic factors such as microRNAs [58,59]. The present study shows that the interaction between BRG1 and p300 increases during the transition from the LVH stage to the HF stage, as does the recruitment of BRG1 onto the hypertrophy-responsive gene promoter. It has been reported that, in cardiomyocytes, p300 autoacetylation is promoted by oxidative stress and is stabilized by increasing resistance to proteasome degradation, resulting in increased HAT activity [60]. It is possible that the bromodomain of BRG1 directly binds to the autoacetylation sites of p300 in HF. Alternatively, as p300 and BRG1 bind to various proteins and form a large complex [61,62], it is possible that BRG1 and p300 regulate transcription by binding indirectly via other proteins in the HF stage.

We have developed the following hypothesis regarding the epigenetic regulatory mechanisms involved in the development of HF (Figure 6). First, p300 is activated due to hypertrophic stresses on the heart such as hypertension. Activated p300 acetylates histone tail domains such as H3K9, resulting in chromatin remodeling. The alteration of chromatin structure that occurs during this remodeling enhances the transcription levels of hypertrophic response genes, leading to LVH. Second, persistent hypertrophic stress induces an increase in the formation of p300/BRG1 complexes, and BRG1 is recruited onto histones. BRG1 then creates a gap between each histone and the DNA wrapped around that histone, inserting p300 into the nucleosome and acetylating H3K122. H3K122 acetylation further reduces DNA-histone affinity, relaxing the nucleosome structure. This relaxation enables the histones to slide along the DNA, opening spaces between them. Finally, large complexes containing transcription factors are recruited into these spaces, thereby significantly activating the gene transcription involved in the HF stage.

In conclusion, the present study suggests that the acetylation of the histone globular domain H3K122 is enhanced during the transition from cardiac hypertrophy to heart failure and that the formation of the p300/BRG1 complex is involved in the acetylation of H3K122. These findings may contribute to the discovery of a novel epigenetic regulation mechanism of heart failure, which may, in turn, lead to a much-needed new treatment for the disease.

## 4. Materials and Methods

### 4.1. Animals

All animal experiments complied with the guidelines on animal experiments of the University of Shizuoka and the Kyoto Medical Center and were performed in accordance with protocols approved by the ethics committees of the two institutions. For this study, 1- to 2-day-old SD rats and C57/BL6J mice were purchased from Japan SLC Inc. (Shizuoka, Japan). Transgenic (TG) mice overexpressing p300 (α-MHC-p300 TG) in the heart were produced in a previous study [9]. Male Dahl salt-sensitive (DS) and salt-resistant (DR) rats were purchased from Japan SLC Inc. The rats were fed an 8% NaCl (high-salt) diet from 6 weeks of age [63,64]. The animals were sacrificed at the LVH stage (12 weeks) and the HF stage (21 weeks). Normotensive DR rats were used as age-matched control animals.

### 4.2. Neonatal Rat Ventricular Cardiomyocyte Culture

Primary cultures of neonatal rat cardiomyocytes were isolated and prepared as described previously [8]. In brief, isolated cardiomyocytes were maintained with D-MEM (Sigma-Aldrich, St. Louis, Missouri) supplemented with 10% FBS (Sigma-Aldrich) and Penicillin-Streptomycin Mixed Solution (Stabilized) (Nacalai Tesque, Kyoto, Japan) in a 37 °C incubator with 5% CO_2_ for 48 h, the cells were stimulated with 30 μM phenylephrine (PE) (Fujifilm Wako Pure Chemical Corporation, Osaka, Japan) for 48 h in an incubator at 37 °C. For the curcumin treatment, the cells were treated with 10 μM curcumin (Sigma-Aldrich) in serum-free DMEM for 2 h and then stimulated with PE.

### 4.3. Echocardiography

Cardiac function was noninvasively evaluated by echocardiography using a 10–12 MHz probe and a Sonos 5500 Ultrasound System according to a method described previously [65]. LVPWT, interventricular septum thickness at end diastole (IVSd), LV internal diameter at end diastole (LVIDd), LV internal diameter at end systole (LVIDs), and FS were measured with M-mode tracing from the short-axis view of the LV at the papillary muscle level. All measurements were performed in a blinded manner according to the guidelines of the American Society for Echocardiology and averaged over 3 consecutive cardiac cycles.

### 4.4. Transfection

Cardiomyocytes were transfected with pCMV-p300 CHA or pcDNA null using Lipofectamine Plus (Invitrogen, Carlsbad, California) as described previously [66]. Small interfering RNA (siRNA) was transfected to the cardiomyocytes with Lipofectamine RNAiMAX (Invitrogen) according to the manufacturer’s instructions, as described previously [65]. In brief, two types of siRNA-p300 (Sigma-Aldrich) were mixed and used in this experiment. Mission^®^ siRNA Universal Negative Control #1 (Sigma-Aldrich) was used as the negative control siRNA. The sequences of si-p300 were constructed as follows. si-p300-1: sense, 5’-CUAGAGACACCUUGUAGUATT-3’; anti-sense, 5’-UACUACAAGGUGUCUCUAGTT-3’. si-p300-2: sense, 5’-GCAUAAAGAGGUCUUCUUUTT-3’; anti-sense, 5’-AAAGAAGACCUCUUUAUGCTT-3’.

### 4.5. Immunoblot Analysis of Histones

Histone extracts were prepared from neonatal rat cardiomyocytes, Dahl-rats, and α-MHC-p300 TG mice as described previously [66]. For Western blotting, rabbit monoclonal anti-histone-H3 antibody, rabbit polyclonal anti-acetyl-histone-H3 (K9) antibody, and rabbit polyclonal anti-acetyl-H3 (K122) antibody (Abcam, Cambridge, UK) were used. The signals were detected with a C-DiGit Chemiluminescent Western Blot Scanner (LI-COR, Lincoln, Nebraska) and a LAS-1000 Plus Luminescent Image Analyzer (Fujifilm, Tokyo, Japan) and quantified using Image Studio LITE software (LI-COR).

### 4.6. Immunoprecipitation and Western Blotting

Nuclear extracts were prepared from cardiomyocytes and rat or mouse hearts, and immunoprecipitation and Western blots were performed as described previously [65]. For immunoprecipitation, mouse monoclonal anti-BRG1 antibody was purchased from Sigma-Aldrich, and normal mouse IgG from Jackson ImmunoResearch Laboratories (West Grove, Pennsylvania). For Western blotting, mouse monoclonal anti-HA probe antibody, rabbit polyclonal anti-p300 polyclonal antibody, and rabbit polyclonal anti-BRG1 antibody were purchased from Abcam and mouse monoclonal anti-β-actin antibody from Sigma-Aldrich. For the analysis of the total amount of BRG1 after immunoprecipitation, the membrane was reprobed with goat polyclonal anti-BRG1 antibody (Santa Cruz Biotechnology, Dallas, TX, USA).

### 4.7. Quantitative RT-PCR

Quantitative RT-PCR assays were performed as previously described [66]. In brief, total RNA from cultured cardiomyocytes and rat hearts was extracted with TRI Reagent (Nacalai Tesque), and cDNA was synthesized using ReverTra Ace^®^ qPCR RT Master Mix (Toyobo, Osaka, Japan) according to the manufacturers’ instructions. Quantified PCR was performed using KOD SYBR qPCR Mix (Toyobo) on a 7500 Real-Time PCR System (Applied Biosystems, Foster City, California). Each mRNA level was normalized to GAPDH. Primers were designed as follows; ANF: 5’-AGGCCATATTGGAGCAAATC-3’, 5’-CATCTTCTCCTCCAGGTGGT-3’. BNP: 5’-ATCTGCCCTCTTGAAAAGCA-3’, 5’-TCGAGCAGATTTGGCTGTTA-3’. β-MHC: 5’-ATCACCAACAACCCCTACGA-3’, 5’-GCGCCTGTCAGCTTGTAAAT-3’. GAPDH: 5’-TGGTGAAGGTCGGTGTGAAC-3’, 5’-GTTGAACTTGCCGTGGGTAG-3’.

### 4.8. Immunofluorescence Staining and Measurement of Surface Area of Cardiomyocytes

Immunofluorescence staining was carried out with antimyosin heavy chain antibody (Leica Biosystems, Wetzlar, Germany) and Alexa555-conjuagted antimouse IgG antibody (Invitrogen) as previously described [65]. A total of 50 myocardial cells were randomly selected, and the surface area of these cells was measured with ImageJ v4.16 software. Detection of nuclear acetylation was done by immunofluorescence. Staining for acetylated histone was performed using the immunofluorescence method previously described [66], with some modifications. In brief, cardiomyocytes were fixed with 3.7% formaldehyde for 15 min, blocked in 1% BSA and 0.5% NP-40/TBS-Ca for 1 h, and then incubated overnight with anti-acetyl-histone-H3 (K9) antibody or anti-acetyl-histone-H3 (K122) antibody with anti-myosin heavy chain antibody. Further, the cells were incubated with antirabbit Cy3 antibody and antimouse DyLight-649 antibody for 2 h and then stained with Hoechst 33432 (Nacalai Tesque). Immunofluorescence was observed using a LSM 510 Laser Scanning Microscope (Carl Zeiss, Oberkochen, Germany).

### 4.9. Chromatin Immunoprecipitation Assay

ChIP assay was performed as described previously, with the following modifications [66]. In brief, primary cardiomyocytes prepared from neonatal rats were treated with or without 30 μM PE for 15, 60, or 240 min. After fixation of the genomic DNA and nuclear proteins with formalin, the cellular extract was sonicated with a Bioruptor UCD-250 sonicator (Cosmo Bio, Tokyo, Japan); the DNA-protein complex was immunoprecipitated with anti-acetyl-histone-H3 (K9) antibody, anti-acetyl-histone-H3 (K122) antibody, or rabbit IgG; and the immunocomplexes were collected by incubation with antirabbit IgG-conjugated Dynabeads (Invitrogen). 

For the in vivo ChIP assay, frozen hearts were chopped into small pieces (between 1 and 3 mm^3^), then these tissues were transferred into a tube with PBS (containing protease inhibitors) and 1.5 % (v/v) formaldehyde was added on ice. Then, the fixed cells were sonicated. Next, chromatin was incubated with anti-acetyl-histone-H3 (K9) antibody, anti-acetyl-histone-H3 (K122) antibody, rabbit polyclonal anti-p300 polyclonal antibody, rabbit polyclonal anti-BRG1 antibody (Abcam), or rabbit IgG as a control and then precipitated with antirabbit IgG-conjugated Dynabeads (Invitrogen). Precipitated DNA was purified using phenol:chloroform:isoamyl alcohol solution and quantified using KOD SYBR qPCR Mix (Toyoba) on a 7500 Real-Time PCR System (Applied Biosystems). Primers were designed as follows; ANF upstream (-2177 bp to -2014 bp): Fw 5’-TGTGTTTGCTTGTGCTAGGCCC-3’, Rv 5’-TAAGTGGGCTGGTATGTGCTTG-3’. BNP upstream (-2497 bp to -2259 bp): Fw 5’-CACCAAGCCACACTCTGAAG-3’, Rv 5’-TGGCTGAAGATTGAATGCAG-3’. β-MHC upstream (-3364 bp to -3176 bp): 5’-GCAGTCTGGATCCCTGATGT-3’, Rv 5’-GACACTGGGGCACAGAGATT-3’. 

### 4.10. Statistical Analysis

The results are expressed as the means ± SE. Statistical comparisons between experimental groups were performed using one-way or two-way ANOVA followed by Tukey test and unpaired *t*-test. The results were considered significant if *p* was <0.05.

## Figures and Tables

**Figure 1 ijms-22-01771-f001:**
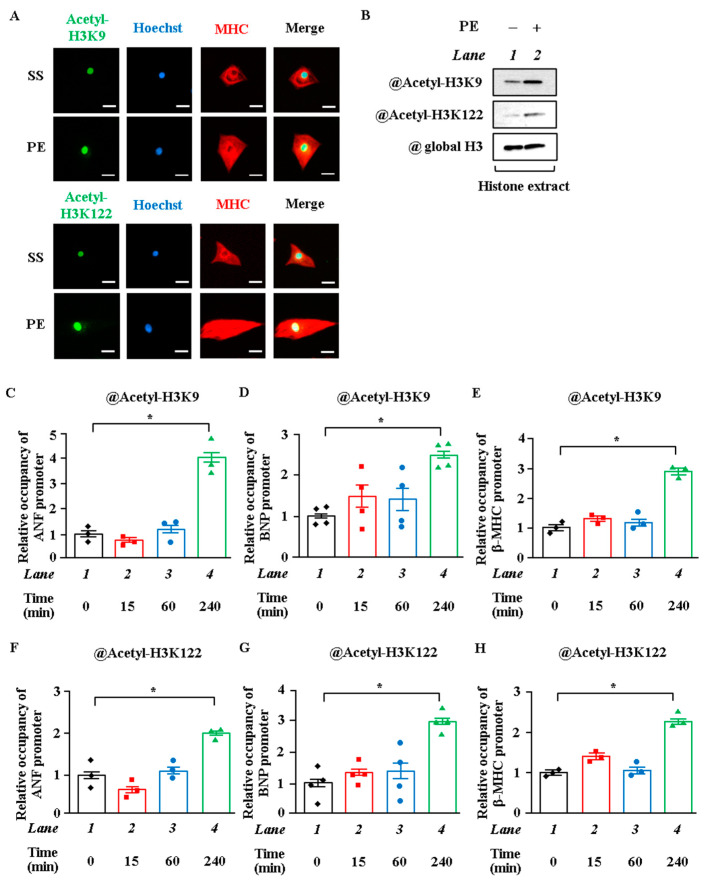
Phenylephrine stimulation induced histone acetylation in cardiomyocyte hypertrophy. (**A**) Primary cultured neonatal rat cardiomyocytes treated with or without phenylephrine (PE) (30 μM) for 48 h. Immunofluorescence staining was performed with anti-acetyl-histone H3K9, anti-acetyl-histone H3K122, and anti-MHC antibody. Green: Acetyl-H3K9 or Acetyl-H3K122. Blue: Hoechst. Red: MHC. Scale bar: 20 μm. (**B**) Histone was extracted with hydrochloric acid from cardiomyocytes treated with or without PE (30 μM) for 48 h. Western blotting was performed with anti-acetyl-histone H3K9 antibody, anti-acetyl-histone H3K122 antibody, and anti-histone H3 antibody. (**C**–**H**) ChIP assays were performed using cardiomyocyte lysates treated with or without PE for 0, 15, 60, or 240 min with anti-acetyl-histone H3K9 antibody (**C**–**E**), anti-acetyl-histone H3K122 antibody (**F**–**H**), or normal rabbit IgG as a negative control (not detected). N = 3 to 4; *one-way* ANOVA followed by Tukey test. * *p* < 0.05.

**Figure 2 ijms-22-01771-f002:**
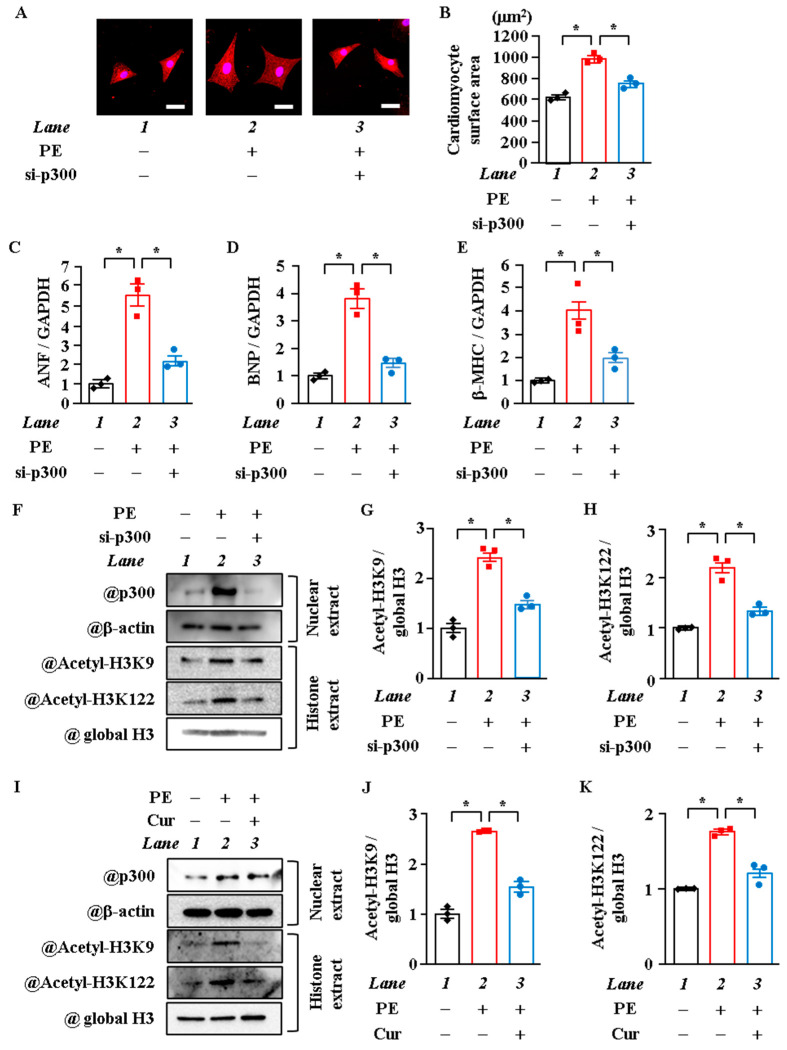
Knockdown of p300 and p300-HAT inhibitor suppressed histone acetylation. Primary cultured neonatal rat cardiomyocytes were transfected with p300 si-RNA or si-control as a control (50 nM, respectively). (**A** and **B**) Immunostaining was performed on primary cultured cardiomyocytes with anti-MHC antibody. The areas of 50 randomly chosen cells were measured using ImageJ v4.16. (**A**) is a photograph of representative cardiomyocytes, and (**B**) is a quantification of (**A**). Scale bar: 20 μm. (**C**–**D**) mRNA was extracted from the cardiomyocytes, and mRNA levels of ANF (**C**), BNP (**D**), and β-MHC (**E**) were measured by qRT-PCR assay. (**F**–**H**) Nuclear proteins (nuclear extract) and histone fraction (histone extract) were extracted from the cardiomyocytes. (**F**) is a photograph of a representative Western blot, and (**G** and **H**) are quantifications of (**F**). (**I**–**K**) Primary cultured cardiomyocytes were treated with 10 μM of curcumin (Cur), a p300 specific HAT inhibitor, and were stimulated with 30 μM of PE. Nuclear proteins and histone fraction were extracted from primary cardiomyocytes after 48 h of PE stimulation. (I) is a photograph of a representative Western blot, and (**J**) and (**K**) are quantifications of (**I**). (**B**–**E**,**G**,**H**,**J**, and **K**), N = 3; *one-way* ANOVA followed by Tukey test. * *p* < 0.05.

**Figure 3 ijms-22-01771-f003:**
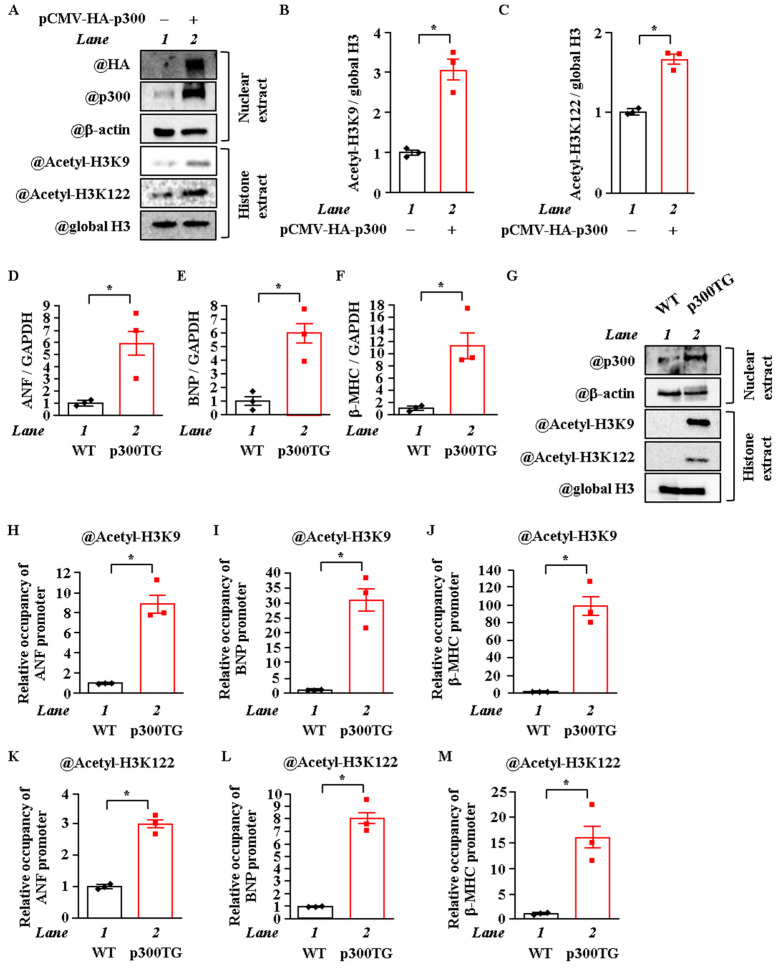
Cardiac overexpression of p300 increased the acetylation of H3K9 and H3K122 in vitro and in vivo. (**A**–**C**) Cardiomyocytes were transfected with pCMV-p300 or pcDNA null vector. Nuclear proteins (nuclear extract) and histone fraction (histone extract) were extracted from these cardiomyocytes. (**A**) is a photograph of a representative Western blot, and (**B**) and (**C**) are quantifications of (**A**). (**D**–**F**) mRNA was extracted from the hearts of p300 TG or WT mice at 26 weeks of age, and the mRNA levels of ANF (**D**), BNP (**E**), and β-MHC (**F**) were measured. (**G**) Nuclear proteins and histone fraction were extracted from hearts of TG and WT mice. (**H**–**M**) In vivo ChIP assay was performed using heart lysates from p300 TG and WT mice with anti-acetyl-histone H3K9 antibody (**H**–**J**), anti-acetyl-histone H3K122 antibody (**K**–**M**), or normal rabbit IgG as a negative control (not detected). (**B**–**F** and **H**–**M**), N = 3; unpaired *t*-test. * *p* < 0.05.

**Figure 4 ijms-22-01771-f004:**
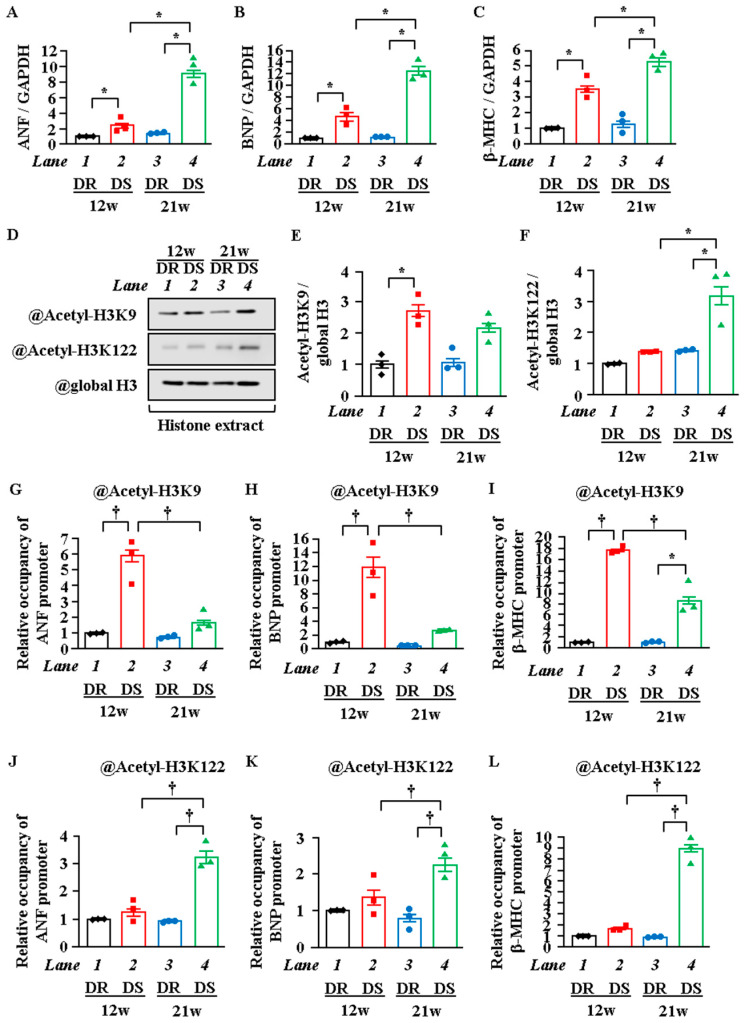
H3K9 acetylation was increased at the LVH stage, but H3K122 acetylation was increased at the HF stage. At 6 weeks of age, male Dahl salt-sensitive (DS) and salt-resistant (DR) rats were given an 8% NaCl diet for the following 6 or 15 weeks. Using echocardiography, left ventricular hypertrophy (LVH) stage was confirmed at 12 weeks of age, and heart failure (HF) stage was confirmed at 21 weeks of age. (**A**–**C**) mRNA was extracted from the hearts of DR and DS rats at 12 or 21 weeks of age, and mRNA levels of ANF (**A**), BNP (**B**), and β-MHC (**C**) were measured. (**D**–**F**) Nuclear proteins and histone fraction were extracted from the hearts of DR and DS rats at 12 or 21 weeks of age. (**D**) is a photograph of a representative Western blot, and (**E**,**F**) are quantifications (**D**). (**G**–**L**) In vivo ChIP assay was performed using heart lysates of DR and DS rats at 12 or 21 weeks of age with anti-acetyl-histone H3K9 antibody (**G**–**I**), anti-acetyl-histone H3K122 antibody (**J**–**L**), or normal rabbit IgG as a negative control (not detected). (**A**–**C** and **E**–**L**), N = 3; two-way ANOVA followed by Tukey test. * *p* < 0.05, † *p* < 0.01.

**Figure 5 ijms-22-01771-f005:**
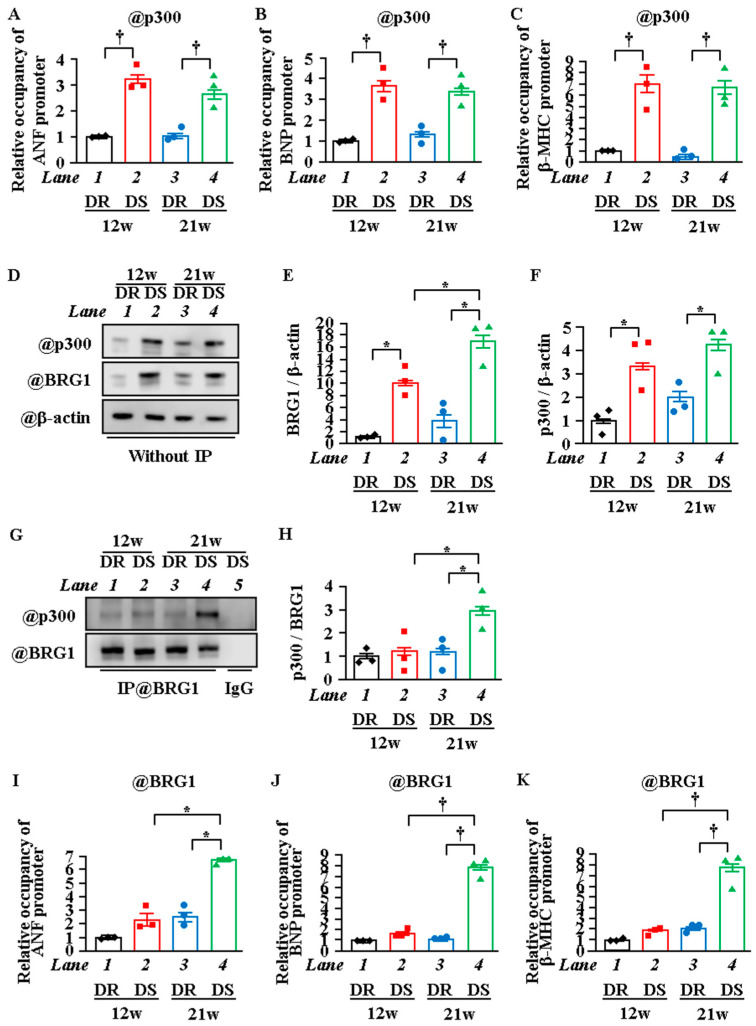
The interaction between p300 and BRG1 was increased in the HF stage. (**A**–**C**) In vivo ChIP assay was performed using the heart lysates of DR and DS rats at 12 or 21 weeks of age with anti-p300 antibody (**A**–**C**) or normal rabbit IgG as a negative control (not detected). (**D**–**F**) Nuclear proteins were extracted from the hearts of DR or DS rats at 12 or 21 weeks of age. (**D**) is a photograph of a representative Western blot, and (**E** and **F**) are a quantification of (**D**). (**G**,**H**) Dahl rat heart lysates were used for immunoprecipitation with either control rabbit IgG or BRG1 antibody. Immunoblots were performed for p300 and BRG1. (**G**) is a photograph of a representative Western blot, and (**H**) is a quantification of (**G**). (**I**–**K**) In vivo ChIP assay was performed using heart lysates of DR or DS rats at 12 or 21 weeks of age with anti-BRG1 antibody or normal rabbit IgG as a negative control (not detected). (**A**–**C**,**E**,**F**,**H**–**K**), N = 3; *two-way* ANOVA followed by Tukey test. * *p* < 0.05, † *p* < 0.01.

**Figure 6 ijms-22-01771-f006:**
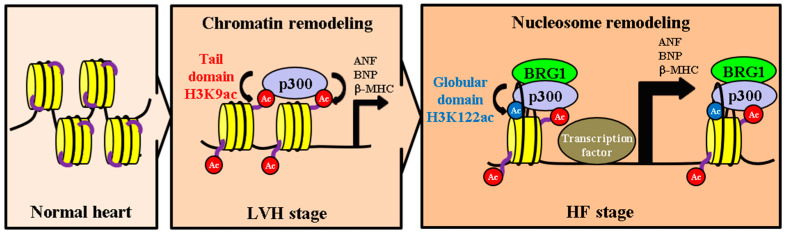
Model of histone acetylation during the transition from LVH to HF. LVH: left ventricular hypertrophy. HF: heart failure. Red Ac: H3K9 acetylation. Blue Ac: H3K122 acetylation. Purple lines: histone tail domains. Black lines: DNA.

## Data Availability

The data presented in this study are available on request from the corresponding author.

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
