# Peer review of "Histone Acetylation Domains Are Differentially Induced during Development of Heart Failure in Dahl Salt-Sensitive Rats"

_ijms, 2021, doi:10.3390/ijms22041771_

Round 1

Reviewer 1 Report

Funamoto et al investigated epigenetic regulation of histone acetylation in relation to disease stages. Experiments were conducted on neonate cardiomyocytes stimulated with PE, and effect of HAC p300 was addressed by p300 KD or inhibitors. In vivo experiments were conducted on p300-TG mice and more importantly on Dahl salt-sensitive rats (DS) at hypertrophic and HF stages, as confirmed by echocardiography. WB, RT-PCR, IHC and ChiP assay were applied. The authors concluded that histone acetylation at H3K9 and H3K122, regulated by p300/BRG1, are critical in controlling expression of a group of hypertrophy-related genes and hence the extent of myocardial hypertrophy, and that increased acetyl-H3K122 is associated with decompensation into HF. The quality of data appears to be good and methodologies were well validated and solid. The way of data presentation and discussion is clear. Demonstration of epigenetic signatures associated with hypertrophy or decompensation into HF, respectively, is important. In this context, revealing increased acetyl-H3K122 in failing heart bears potential therapeutic implication.

I have only one comment for the authors to consider in revision.

Your statement in Discussion lines 307-308 “Previously, nothing was known about changes in the binding between p300 and BRG1, or about the coordinated regulation of the BRG1/p300 complex in the LVH and HF stages” is not correct. El-Osta’s group has conducted a series of studies using the same TAC model and illustrated the in the hypertrophic stage, epigenetic mechanism of MHC isoform switch that involves Swi/Snf (Brm, BRG1) interacting with HADAC2 and p300 [Chang L, et al: Epigenetics 2011;6:760]. More recent work from this group suggests that such reprograming is also influenced by miR-208b and Ezh2 as a polycomb-group protein [Mathiyalagan P, et al: Nucleic Acids Res 2014,42:790]. Some of these works very relevant to this paper should be referred to and discussed. These works were also reviewed [Cardiovasc Res 2014, 103:7-16]. Having said this, their works were concentrated on epigenetic mechanism for hypertrophic gene profile, but not on HF.

Please refer to these works that are directly relevant to your findings in the present paper, and revise your discussion.

Author Response

February 6, 2021

Dear Reviewers,

Thank you very much for your valuable comments on our manuscript. We have made revisions according to your suggestions. We have given careful attention to each of your comments and respond as follows:

Reviewer 1

Q : Your statement in Discussion lines 307-308 “Previously, nothing was known about changes in the binding between p300 and BRG1, or about the coordinated regulation of the BRG1/p300 complex in the LVH and HF stages” is not correct. El-Osta’s group has conducted a series of studies using the same TAC model and illustrated the in the hypertrophic stage, epigenetic mechanism of MHC isoform switch that involves Swi/Snf (Brm, BRG1) interacting with HADAC2 and p300 [Chang L, et al: Epigenetics 2011;6:760]. More recent work from this group suggests that such reprograming is also influenced by miR-208b and Ezh2 as a polycomb-group protein [Mathiyalagan P, et al: Nucleic Acids Res 2014,42:790]. Some of these works very relevant to this paper should be referred to and discussed. These works were also reviewed [Cardiovasc Res 2014, 103:7-16]. Having said this, their works were concentrated on epigenetic mechanism for hypertrophic gene profile, but not on HF.

A : Thank you very much for your valuable advice. In response to your comment, we have changed the revised manuscript as follows:

Line310-315

“In cardiac hypertrophy, it has been reported that the interaction of the Swi/Snf (Brm, BRG1) complex with HADAC2 and p300 is associated with the MHC isoform switch [57]. In addition, this switch has been shown to be regulated by epigenetic factors such as microRNAs [58, 59]. The present study shows that the interaction between BRG1 and p300 increases during the transition from the LVH stage to the HF stage, as does the recruitment of BRG1 onto the hypertrophy-responsive gene promoter.”

  1. Chang, L.; Kiriazis, H.; Gao, X. M.; Du, X. J.; Osta, A. E. Cardiac genes show contextual SWI/SNF interactions with distinguishable gene activities. Epigenetics 2011, 6, 760-768.
  2. Mathiyalagan, P.; Okabe,J.; Chang,L.; Su,Y.; Du,X.J.; Osta, A.E. The primary microRNA-208b interacts with Polycomb-group protein, Ezh2, to regulate gene expression in the heart. Nucleic Acids Res 2014, 42, 790–803.
  3. Mathiyalagan, P.; Keating, S.T.; Du, X. J.; Osta, A.E. Chromatin modifications remodel cardiac gene expression. Cardiovasc Res 2014,103, 7-16.

Reviewer 2 Report

I read the report by Funamoto et al with great interest. Cardiac hypertrophy and subsequent heart failure remain a massive health burden and research in the area of epigenetic drugs is needed not only to identify new lead compounds for future exploration but also to understand the underlying mechanisms. Funamoto and colleagues characterise an interaction between p300 and Brg1 which co-occurs with H3K122ac and is involved in the progression of hypertrophy to heart failure. I am happy to recommend acceptance if the following can be addressed.

1. L102/L117 Please state in the legend of Figure 1 what type of cardiomyocytes are being cultured. Are these SD rats or C57 mice? 

2. L114 (Data not shown) is not really acceptable anymore. Please supply the supporting data as a supplement.

3. Figure 2F: The p300 bands are not clear and this should be repeated to confirm that p300 is actually knocked-down. 

4. Methods section should describe cell culture experiments before echocardiography.

5. Authors assert on line 32 that acetylation is the main type of histone modification but histone methylation could be argued to be just as abundant and important. 

Author Response

February 6, 2021

Dear Reviewers,

Thank you very much for your valuable comments on our manuscript. We have made revisions according to your suggestions. We have given careful attention to each of your comments and respond as follows:

Reviewer 2

Q1 : L102/L117 Please state in the legend of Figure 1 what type of cardiomyocytes are being cultured. Are these SD rats or C57 mice?

A : Thank you very much for your valuable advice. As you suggested, we have revised the sentences in the revised manuscript as follow:

Line119-120

“Primary cultured neonatal rat cardiomyocytes treated with or without phenylephrine (PE) (30 μM) for 48 h.”

Line148-150

“Primary cultured neonatal rat cardiomyocytes were transfected with p300 si-RNA or si-control as a control (50 nM, respectively). ”

Q2 : L114 (Data not shown) is not really acceptable anymore. Please supply the supporting data as a supplement.

A : Thank you very much for your helpful comment. As you suggested, we have added the ChIP assay data as a supplement. Accordingly, we have added the data in the revised manuscript as follows:

Line 114-116

“On the other hand, the acetylation levels of H3K9 and H3K122 around the upstream region of these promoters were not changed after PE stimulation (Figures S1A-S1F).”

Supplemental Fig. 1 The acetylation levels of H3K9 and H3K122 were not changed around the upstream region of the hypertrophic response gene promoters

(A-F) ChIP assays were performed using cardiomyocyte lysates treated with or without PE for 0, 15, 60, or 240 min with anti-acetyl-histone H3K9 antibody (A-C), anti-acetyl-histone H3K122 antibody (D-F), or normal rabbit IgG as a negative control (not detected). N=3 to 4; one-way ANOVA followed by Tukey test. * p < 0.05.

Line460-465

“The primers were designed as follows. ANF upstream (-2177 bp to -2014 bp): Fw 5’-TGTGTTTGCTTGTGCTAGGCCC-3’, Rv 5’-TAAGTGGGCTGGTATGTGCTTG-3’. BNP upstream (-2497 bp to -2259 bp): Fw 5’-CACCAAGCCACACTCTGAAG-3’, Rv 5’-TGGCTGAAGATTGAATGCAG-3’. b-MHC upstream (-3364 bp to -3176 bp): 5’-GCAGTCTGGATCCCTGATGT-3’, Rv 5’-GACACTGGGGCACAGAGATT-3’.”

Q3 : Figure 2F: The p300 bands are not clear and this should be repeated to confirm that p300 is actually knocked-down.

A : Thank you very much for your helpful comment. As you suggested, we have updated the data in the revised manuscript as follows:

Q4 : Methods section should describe cell culture experiments before echocardiography.

A : Thank you very much for your valuable advice. In response, we have changed the sentences in the revised manuscript as follows:

Lines 361-377

4.2. Neonatal rat ventricular cardiomyocyte culture

Primary cultures of neonatal rat cardiomyocytes were isolated and prepared as described previously [8]. In brief, isolated cardiomyocytes were maintained with D-MEM (Sigma-Aldrich, St. Louis, Missouri) supplemented with 10% FBS (Sigma-Aldrich) and Penicillin-Streptomycin Mixed Solution (Stabilized) (Nacalai Tesque, Kyoto, Japan) in a 37oC incubator with 5% CO2 for 48 hours, the cells were stimulated with 30 μM phenylephrine (PE) (Fujifilm Wako Pure Chemical Corporation, Osaka, Japan) for 48 hours in an incubator at 37oC. For the curcumin treatment, the cells were treated with 10 μM curcumin (Sigma-Aldrich) in serum-free DMEM for 2 hours and then stimulated with PE.

4.3. Echocardiography

Cardiac function was non-invasively evaluated by echocardiography using a 10-12 MHz probe and a Sonos 5500 Ultrasound System according to a method described previously [65]. LVPWT, interventricular septum thickness at end diastole (IVSd), LV internal diameter at end diastole (LVIDd), LV internal diameter at end systole (LVIDs), and FS were measured with M-mode tracing from the short-axis view of the LV at the papillary muscle level. All measurements were performed in a blinded manner according to the guidelines of the American Society for Echocardiology and averaged over 3 consecutive cardiac cycles.”

Q5 : Authors assert on line 32 that acetylation is the main type of histone modification but histone methylation could be argued to be just as abundant and important.

A : Thank you very much for your helpful comment. In response, we have added references regarding histone methylation. We have added the following sentences and references to the revised manuscript:

Lines 31-34

“The main types of post-translational histone modification are acetylation and methylation. These modifications play essential roles in a variety of regulatory mechanisms and are controlled by acetyltransferase or methylase respectively [6].”

  1. Olio, F. D.; Trinchera, M. Epigenetic Bases of Aberrant Glycosylation in Cancer. Int J Mol Sci 2017, 18, 998.